# MACER: Attack-free and Scalable Robust Training via Maximizing Certified Radius

**Runtian Zhai**[1*]**, Chen Dan**[2*]**, Di He**[1*]**, Huan Zhang**[3]**,**
**Boqing Gong**[4]**, Pradeep Ravikumar**[2]**, Cho-Jui Hsieh**[3] **& Liwei Wang**[1]
[1]Peking University  [2]CMU  [3]UCLA  [4]Google  *Equal Contribution

{zhairuntian, di_he}@pku.edu.cn, {cdan, pradeepr}@cs.cmu.edu,
huanzhang@ucla.edu, boqinggo@outlook.com, chohsieh@cs.ucla.edu, wanglw@cis.pku.edu.cn

## Abstract

Adversarial training is one of the most popular ways to learn robust models but is usually attack-dependent and time costly. In this paper, we propose the MACER algorithm, which learns robust models without using adversarial training but performs better than all existing provable $l_2$-defenses. Recent work (Cohen et al., 2019) shows that randomized smoothing can be used to provide a certified $l_2$ radius to smoothed classifiers, and our algorithm trains provably robust smoothed classifiers via MAximizing the CErtified Radius (MACER). The attack-free characteristic makes MACER faster to train and easier to optimize. In our experiments, we show that our method can be applied to modern deep neural networks on a wide range of datasets, including Cifar-10, ImageNet, MNIST, and SVHN. For all tasks, MACER spends less training time than state-of-the-art adversarial training algorithms, and the learned models achieve larger average certified radii. Our code is available at `https://github.com/RuntianZ/macer`.

## 1 Introduction

Modern neural network classifiers are able to achieve very high accuracy on image classification tasks but are sensitive to small, adversarially chosen perturbations to the inputs (Szegedy et al., 2013; Biggio et al., 2013). Given an image $x$ that is correctly classified by a neural network, a malicious attacker may find a small adversarial perturbation $\delta$ such that the perturbed image $x + \delta$, though visually indistinguishable from the original image, is assigned to a wrong class with high confidence by the network. Such vulnerability creates security concerns in many real-world applications.

Researchers have proposed a variety of defense methods to improve the robustness of neural networks. Most of the existing defenses are based on adversarial training (Szegedy et al., 2013; Madry et al., 2017; Goodfellow et al., 2015; Huang et al., 2015; Athalye et al., 2018; Ding et al., 2020). During training, these methods first learn on-the-fly adversarial examples of the inputs with multiple attack iterations and then update model parameters using these perturbed samples together with the original labels. However, such approaches depend on a particular (class of) attack method. It cannot be formally guaranteed whether the resulting model is also robust against other attacks. Moreover, attack iterations are usually quite expensive. As a result, adversarial training runs very slowly.

Another line of algorithms trains robust models by maximizing the certified radius provided by robust certification methods (Weng et al., 2018; Wong & Kolter, 2018; Zhang et al., 2018; Mirman et al., 2018; Wang et al., 2018; Gowal et al., 2018; Zhang et al., 2019c). Using linear or convex relaxations of fully connected ReLU networks, a robust certification method computes a "safe radius" $r$ for a classifier at a given input such that at any point within the neighboring radius-$r$ ball of the input, the classifier is guaranteed to have unchanged predictions. However, the certification methods are usually computationally expensive and can only handle shallow neural networks with ReLU activations, so these training algorithms have troubles in scaling to modern networks.

In this work, we propose an attack-free and scalable method to train robust deep neural networks. We mainly leverage the recent randomized smoothing technique (Cohen et al., 2019). A randomized smoothed classifier $g$ for an arbitrary classifier $f$ is defined as $g(x) = \mathbb{E}_\eta f(x + \eta)$, in which $\eta \sim \mathcal{N}(0, \sigma^2 \boldsymbol{I})$. While Cohen et al. (2019) derived how to analytically compute the certified radius of the randomly smoothed classifier $g$, they did not show how to maximize that radius to make the classifier

$g$ robust. Salman et al. (2019) proposed SmoothAdv to improve the robustness of $g$, but it still relies on the expensive attack iterations. Instead of adversarial training, we propose to learn robust models by directly taking the certified radius into the objective. We outline a few challenging desiderata any practical instantiation of this idea would however have to satisfy, and provide approaches to address each of these in turn. A discussion of these desiderata, as well as a detailed implementation of our approach is provided in Section 4. And as we show both theoretically and empirically, our method is numerically stable and accounts for both classification accuracy and robustness.

Our contributions are summarized as follows:

- We propose an attack-free and scalable robust training algorithm by MAximizing the CErtified Radius (MACER). MACER has the following advantages compared to previous works:
  - Different from adversarial training, we train robust models by directly maximizing the certified radius without specifying any attack strategies, and the learned model can achieve provable robustness against any possible attack in the certified region. Additionally, by avoiding time-consuming attack iterations, our proposed algorithm runs much faster than adversarial training.
  - Different from other methods (Wong & Kolter, 2018) that maximize the certified radius but are not scalable to deep neural networks, our method can be applied to architectures of any size. This makes our algorithm more practical in real scenarios.
- We empirically evaluate our proposed method through extensive experiments on Cifar-10, ImageNet, MNIST, and SVHN. On all tasks, MACER achieves better performance than state-of-the-art algorithms. MACER is also exceptionally fast. For example, on ImageNet, MACER uses 39% less training time than adversarial training but still performs better.

## 2 RELATED WORK

Neural networks trained by standard SGD are not robust – a small and human imperceptible perturbation can easily change the prediction of a network. In the white-box setting, methods have been proposed to construct adversarial examples with small $\ell_\infty$ or $\ell_2$ perturbations (Goodfellow et al., 2015; Madry et al., 2017; Carlini & Wagner, 2016; Moosavi-Dezfooli et al., 2015). Furthermore, even in the black-box setting where the adversary does not have access to the model structure and parameters, adversarial examples can be found by either transfer attack (Papernot et al., 2016) or optimization-based approaches (Chen et al., 2017; Rauber et al., 2017; Cheng et al., 2019). It is thus important to study how to improve the robustness of neural networks against adversarial examples.

**Adversarial training** So far, adversarial training has been the most successful robust training method according to many recent studies. Adversarial training was first proposed in Szegedy et al. (2013) and Goodfellow et al. (2015), where they showed that adding adversarial examples to the training set can improve the robustness against such attacks. More recently, Madry et al. (2017) formulated adversarial training as a min-max optimization problem and demonstrated that adversarial training with PGD attack leads to empirical robust models. Zhang et al. (2019b) further decomposed the robust error as the sum of natural error and boundary error for better performance. Finally, Gao et al. (2019) proved the convergence of adversarial training. Although models obtained by adversarial training empirically achieve good performance, they do not have certified error guarantees.

Despite the popularity of PGD-based adversarial training, one major issue is that its speed is too slow. Some recent papers propose methods to accelerate adversarial training. For example, Free-m (Shafahi et al., 2019) replays an adversarial example several times in one iteration, YOPO-m-n (Zhang et al., 2019a) restricts back propagation in PGD within the first layer, and Qin et al. (2019) estimates the adversary with local linearization.

**Robustness certification and provable defense** Many defense algorithms proposed in the past few years were claimed to be effective, but Athalye et al. (2018) showed that most of them are based on "gradient masking" and can be bypassed by more carefully designed attacks. It is thus important to study how to measure the provable robustness of a network. A robustness certification algorithm takes a classifier $f$ and an input point $x$ as inputs, and outputs a "safe radius" $r$ such that for any $\delta$ subject to $\|\delta\| \leq r$, $f(x) = f(x + \delta)$. Several algorithms have been proposed recently, including the convex polytope technique (Wong & Kolter, 2018), abstract interpretation methods (Singh et al., 2018; Gehr et al., 2018) and the recursive propagation algrithms (Weng et al., 2018; Zhang et al.,

2018). These methods can provide attack-agnostic robust error lower bounds. Moreover, to achieve networks with nontrivial *certified robust error*, one can train a network by minimizing the certified robust error computed by the above-mentioned methods, and several algorithms have been proposed in the past year (Wong & Kolter, 2018; Wong et al., 2018; Wang et al., 2018; Gowal et al., 2018; Zhang et al., 2019c; Mirman et al., 2018). Unfortunately, they can only be applied to shallow networks with limited activation and run very slowly.

More recently, researchers found a new class of certification methods called randomized smoothing. The idea of randomization has been used for defense in several previous works (Xie et al., 2017; Liu et al., 2018) but without any certification. Later on, Lecuyer et al. (2018) first showed that if a Gaussian random noise is added to the input or any intermediate layer. A certified guarantee on small $\ell_2$ perturbation can be computed via differential privacy. Li et al. (2018) and Cohen et al. (2019) then provided improved ways to compute the $\ell_2$ certified robust error for Gaussian smoothed models. In this paper, we propose a new algorithm to train on these $\ell_2$ certified error bounds to significantly reduce the certified error and achieve better provable adversarial robustness.

## 3 PRELIMINARIES

**Problem setup**  Consider a standard classification task with an underlying data distribution $p_{\text{data}}$ over pairs of examples $x \in \mathcal{X} \subset \mathbb{R}^d$ and corresponding labels $y \in \mathcal{Y} = \{1, 2, \cdots, K\}$. Usually $p_{\text{data}}$ is unknown and we can only access a training set $\mathcal{S} = \{(x_1, y_1), \cdots, (x_n, y_n)\}$ in which $(x_i, y_i)$ is *i.i.d.* drawn from $p_{\text{data}}$, $i = 1, 2, \cdots, n$. The empirical data distribution (uniform distribution over $\mathcal{S}$) is denoted by $\hat{p}_{\text{data}}$. Let $f \in \mathcal{F}$ be the classifier of interest that maps any $x \in \mathcal{X}$ to $\mathcal{Y}$. Usually $f$ is parameterized by a set of parameters $\theta$, so we also write it as $f_\theta$.

We call $x' = x + \delta$ an *adversarial example* of $x$ to classifier $f_\theta$ if $f_\theta$ can correctly classify $x$ but assigns a different label to $x'$. Following many previous works (Cohen et al., 2019; Salman et al., 2019), we focus on the setting where $\delta$ satisfies $\ell_2$ norm constraint $\|\delta\|_2 \leq \epsilon$. We say that the model $f_\theta$ is $l_2^\epsilon$-*robust* at $(x, y)$ if it correctly classifies $x$ as $y$ and for any $\|\delta\|_2 \leq \epsilon$, the model classifies $x + \delta$ as $y$. In the problem of robust classification, our ultimate goal is to find a model that is $l_2^\epsilon$-robust at $(x, y)$ with high probability over $(x, y) \sim p_{\text{data}}$ for a given $\epsilon > 0$.

**Neural network**  In image classification we often use deep neural networks. Let $u_\theta : \mathcal{X} \to \mathbb{R}^K$ be a neural network, whose output at input $x$ is a vector $(u_\theta^1(x), ..., u_\theta^K(x))$. The classifier induced by $u_\theta(x)$ is $f_\theta(x) = \arg\max_{c \in \mathcal{Y}} u_\theta^c(x)$.

In order to train $\theta$ by minimizing a loss function such as cross entropy, we always use a softmax layer on $u_\theta$ to normalize it into a probability distribution. The resulting network is $z_\theta(\cdot; \beta) : \mathcal{X} \to \mathcal{P}(K)$[1], which is given by $z_\theta^c(x; \beta) = e^{\beta u_\theta^c(x)} / \sum_{c' \in \mathcal{Y}} e^{\beta u_\theta^{c'}(x)}, \forall c \in \mathcal{Y}$, $\beta$ is the inverse temperature. For simplicity, we will use $z_\theta(x)$ to refer to $z_\theta(x; \beta)$ when the meaning is clear from context. The vector $z_\theta(x) = (z_\theta^1(x), \cdots, z_\theta^K(x))$ is commonly regarded as the "likelihood vector", and $z_\theta^c(x)$ measures how likely input $x$ belongs to class $c$.

**Robust radius**  By definition, the $l_2^\epsilon$-robustness of $f_\theta$ at a data point $(x, y)$ depends on the radius of the largest $l_2$ ball centered at $x$ in which $f_\theta$ does not change its prediction. This radius is called the *robust radius*, which is formally defined as

$$R(f_\theta; x, y) = \begin{cases} \inf_{f_\theta(x') \neq f_\theta(x)} \|x' - x\|_2, & \text{when } f_\theta(x) = y \\ 0, & \text{when } f_\theta(x) \neq y \end{cases} \tag{1}$$

Recall that our ultimate goal is to train a classifier which is $l_2^\epsilon$-robust at $(x, y)$ with high probability over the sampling of $(x, y) \sim p_{\text{data}}$. Mathematically the goal can be expressed as to minimize the expectation of the *0/1 robust classification error*. The error is defined as

$$l_{\epsilon-\text{robust}}^{0/1}(f_\theta; x, y) := 1 - \mathbf{1}_{\{R(f_\theta; x, y) \geq \epsilon\}}, \tag{2}$$

and the goal is to minimize its expectation over the population

$$L_{\epsilon-\text{robust}}^{0/1}(f_\theta) := \mathbb{E}_{(x, y) \sim p_{\text{data}}} l_{\epsilon-\text{robust}}^{0/1}(f_\theta; x, y). \tag{3}$$

---

[1]The probability simplex in $\mathbb{R}^K$.

It is thus quite natural to improve model robustness via maximizing the robust radius. Unfortunately, computing the robust radius (1) of a classifier induced by a deep neural network is very difficult. Weng et al. (2018) showed that computing the $l_1$ robust radius of a deep neural network is NP-hard. Although there is no result for the $l_2$ radius yet, it is very likely that computing the $l_2$ robust radius is also NP-hard.

**Certified radius**   Many previous works proposed certification methods that seek to derive a tight lower bound of $R(f_\theta; x, y)$ for neural networks (see Section 2 for related work). We call this lower bound *certified radius* and denote it by $CR(f_\theta; x, y)$. The certified radius satisfies $0 \leq CR(f_\theta; x, y) \leq R(f_\theta; x, y)$ for any $f_\theta, x, y$.

The certified radius leads to a guaranteed upper bound of the 0/1 robust classification error, which is called *0/1 certified robust error*. The 0/1 certified robust error of classifier $f_\theta$ on sample $(x, y)$ is defined as

$$l^{0/1}_{\epsilon-\text{certified}}(f_\theta; x, y) := 1 - \mathbf{1}_{\{CR(f_\theta; x, y) \geq \epsilon\}} \tag{4}$$

i.e. a sample is counted as correct only if the certified radius reaches $\epsilon$. The expectation of certified robust error over $(x, y) \sim p_{\text{data}}$ serves as a performance metric of the provable robustness:

$$L^{0/1}_{\epsilon-\text{certified}}(f_\theta) := \mathbb{E}_{(x,y) \sim p_{\text{data}}} l^{0/1}_{\epsilon-\text{certified}}(f_\theta; x, y) \tag{5}$$

Recall that $CR(f_\theta; x, y)$ is a lower bound of the true robust radius, which immediately implies that $L^{0/1}_{\epsilon-\text{certified}}(f_\theta) \geq L^{0/1}_{\epsilon-\text{robust}}(f_\theta)$. Therefore, a small 0/1 certified robust error leads to a small 0/1 robust classification error.

**Randomized smoothing**   In this work, we use the recent randomized smoothing technique (Cohen et al., 2019), which is scalable to any architectures, to obtain the certified radius of smoothed deep neural networks. The key part of randomized smoothing is to use the smoothed version of $f_\theta$, which is denoted by $g_\theta$, to make predictions. The formulation of $g_\theta$ is defined as follows.

**Definition 1.** *For an arbitrary classifier $f_\theta \in \mathcal{F}$ and $\sigma > 0$, the smoothed classifier $g_\theta$ of $f_\theta$ is defined as*

$$g_\theta(x) = \arg\max_{c \in \mathcal{Y}} P_{\eta \sim \mathcal{N}(0, \sigma^2 \boldsymbol{I})}(f_\theta(x + \eta) = c) \tag{6}$$

In short, the smoothed classifier $g_\theta(x)$ returns the label most likely to be returned by $f_\theta$ when its input is sampled from a Gaussian distribution $\mathcal{N}(x, \sigma^2 \boldsymbol{I})$ centered at $x$. Cohen et al. (2019) proves the following theorem, which provides an analytic form of certified radius:

**Theorem 1.** *(Cohen et al., 2019) Let $f_\theta \in \mathcal{F}$, and $\eta \sim \mathcal{N}(0, \sigma^2 \boldsymbol{I})$. Let the smoothed classifier $g_\theta$ be defined as in (6). Let the ground truth of an input $x$ be $y$. If $g_\theta$ classifies $x$ correctly, i.e.*

$$P_\eta(f_\theta(x + \eta) = y) \geq \max_{y' \neq y} P_\eta(f_\theta(x + \eta) = y') \tag{7}$$

*Then $g_\theta$ is provably robust at $x$, with the certified radius given by*

$$\begin{aligned}
CR(g_\theta; x, y) &= \frac{\sigma}{2} [\Phi^{-1}(P_\eta(f_\theta(x + \eta) = y)) - \Phi^{-1}(\max_{y' \neq y} P_\eta(f_\theta(x + \eta) = y'))] \\
&= \frac{\sigma}{2} [\Phi^{-1}(\mathbb{E}_\eta \mathbf{1}_{\{f_\theta(x+\eta)=y\}}) - \Phi^{-1}(\max_{y' \neq y} \mathbb{E}_\eta \mathbf{1}_{\{f_\theta(x+\eta)=y'\}})]
\end{aligned} \tag{8}$$

*where $\Phi$ is the c.d.f. of the standard Gaussian distribution.*

## 4   ROBUST TRAINING VIA MAXIMIZING THE CERTIFIED RADIUS

As we can see from Theorem 1, the value of the certified radius can be estimated by repeatedly sampling Gaussian noises. More importantly, it can be computed for any deep neural networks. This motivates us to design a training method to maximize the certified radius and learn robust models.

To minimize the 0/1 robust classification error in (3) or the 0/1 certified robust error in (5), many previous works (Zhang et al., 2019b; Zhai et al., 2019) proposed to first decompose the error. Note that a classifier $g_\theta$ has a positive 0/1 certified robust error on sample $(x, y)$ if and only if exactly one of the following two cases happens:

- $g_\theta(x) \neq y$, i.e. the classifier misclassifies $x$.
- $g_\theta(x) = y$, but $CR(g_\theta; x, y) < \epsilon$, i.e. the classifier is correct but not robust enough.

Thus, the 0/1 certified robust error can be decomposed as the sum of two error terms: a 0/1 classification error and a 0/1 robustness error:

$$
\begin{aligned}
l_{\epsilon-\text{certified}}^{0/1}(g_\theta; x, y) &= 1 - \mathbf{1}_{\{CR(g_\theta; x, y) \geq \epsilon\}} \\
&= \underbrace{\mathbf{1}_{\{g_\theta(x) \neq y\}}}_{\text{0/1 Classification Error}} + \underbrace{\mathbf{1}_{\{g_\theta(x) = y, CR(g_\theta; x, y) < \epsilon\}}}_{\text{0/1 Robustness Error}}
\end{aligned} \tag{9}
$$

### 4.1 DESIDERATA FOR OBJECTIVE FUNCTIONS

Minimizing the 0-1 error directly is intractable. A classic method is to minimize a surrogate loss instead. The surrogate loss for the 0/1 classification error is called *classification loss* and denoted by $l_C(g_\theta; x, y)$. The surrogate loss for the 0/1 robustness error is called *robustness loss* and denoted by $l_R(g_\theta; x, y)$. Our final objective function is

$$
l(g_\theta; x, y) = l_C(g_\theta; x, y) + l_R(g_\theta; x, y) \tag{10}
$$

We would like our loss functions $l_C(g_\theta; x, y)$ and $l_R(g_\theta; x, y)$ to satisfy some favorable conditions. These conditions are summarized below as (C1) - (C3):

- **(C1)** (Surrogate condition): Surrogate loss should be an upper bound of the original error function, i.e. $l_C(g_\theta; x, y)$ and $l_R(g_\theta; x, y)$ should be upper bounds of $\mathbf{1}_{\{g_\theta(x) \neq y\}}$ and $\mathbf{1}_{\{g_\theta(x) = y, CR(g_\theta; x, y) < \epsilon\}}$, respectively.
- **(C2)** (Differentiablity): $l_C(g_\theta; x, y)$ and $l_R(g_\theta; x, y)$ should be (sub-)differentiable with respect to $\theta$.
- **(C3)** (Numerical stability): The computation of $l_C(g_\theta; x, y)$ and $l_R(g_\theta; x, y)$ and their (sub-)gradients with respect to $\theta$ should be numerically stable.

The surrogate condition (C1) ensures that $l(g_\theta; x, y)$ itself meets the surrogate condition, i.e.

$$
l(g_\theta; x, y) = l_C(g_\theta; x, y) + l_R(g_\theta; x, y) \geq l_{\epsilon-\text{certified}}^{0/1}(g_\theta; x, y) \tag{11}
$$

Conditions (C2) and (C3) ensure that (10) can be stably minimized with first order methods.

### 4.2 SURROGATE LOSSES (FOR CONDITION C1)

We next discuss choices of the surrogate losses that ensure we satisfy condition (C1). The classification surrogate loss is relatively easy to design. There are many widely used loss functions from which we can choose, and in this work we choose the cross-entropy loss as the classification loss:

$$
\mathbf{1}_{\{g_\theta(x) \neq y\}} \leq l_C(g_\theta; x, y) := l_{CE}(g_\theta(x), y) \tag{12}
$$

For the robustness surrogate loss, we choose the hinge loss on the certified radius:

$$
\begin{aligned}
&\mathbf{1}_{\{g_\theta(x) = y, CR(g_\theta; x, y) < \epsilon\}} \\
&\leq \lambda \cdot \max\{\epsilon + \tilde{\epsilon} - CR(g_\theta; x, y), 0\} \cdot \mathbf{1}_{\{g_\theta(x) = y\}} := l_R(g_\theta; x, y)
\end{aligned} \tag{13}
$$

where $\tilde{\epsilon} > 0$ and $\lambda \geq \frac{1}{\tilde{\epsilon}}$. We use the hinge loss because not only does it satisfy the surrogate condition, but also it is numerically stable, which we will discuss in Section 4.4.

### 4.3 DIFFERENTIABLE CERTIFIED RADIUS VIA SOFT RANDOMIZED SMOOTHING (FOR CONDITION C2)

The classification surrogate loss in (12) is differentiable with respect to $\theta$, but the differentiability of the robustness surrogate loss in (13) requires differentiability of $CR(g_\theta; x, y)$. In this section we

will show that the randomized smoothing certified radius in (8) **does not meet** condition (C2), and accordingly, we will introduce *soft randomized smoothing* to solve this problem.

Whether the certified radius (8) is sub-differentiable with respect to $\theta$ boils down to the differentiablity of $\mathbb{E}_\eta \mathbf{1}_{\{f_\theta(x+\eta)=y\}}$. Theoretically, the expectation is indeed differentiable. However, from a practical point of view, the expectation needs to be estimated by Monte Carlo sampling $\mathbb{E}_\eta \mathbf{1}_{\{f_\theta(x+\eta)=y\}} \approx \frac{1}{k} \sum_{j=1}^{k} \mathbf{1}_{\{f_\theta(x+\eta_j)=y\}}$, where $\eta_j$ is i.i.d Gaussian noise and $k$ is the number of samples. This estimation, which is a sum of indicator functions, is not differentiable. Hence, condition (C2) is still not met from the algorithmic perspective.

To tackle this problem, we leverage soft randomized smoothing (Soft-RS). In contrast to the original version of randomized smoothing (Hard-RS), Soft-RS is applied to a neural network $z_\theta(x)$ whose last layer is softmax. The soft smoothed classifier $\tilde{g}_\theta$ is defined as follows.

**Definition 2.** *For a neural network $z_\theta : \mathcal{X} \to \mathcal{P}(K)$ whose last layer is softmax and $\sigma > 0$, the soft smoothed classifier $\tilde{g}_\theta$ of $z_\theta$ is defined as*

$$\tilde{g}_\theta(x) = \arg\max_{c \in \mathcal{Y}} \mathbb{E}_{\eta \sim \mathcal{N}(0,\sigma^2 I)}[z_\theta^c(x+\eta)] \tag{14}$$

Using Lemma 2 in Salman et al. (2019), we prove the following theorem in Appendix A:

**Theorem 2.** *Let the ground truth of an input $x$ be $y$. If $\tilde{g}_\theta$ classifies $x$ correctly, i.e.*

$$\mathbb{E}_\eta[z_\theta^y(x+\eta)] \geq \max_{y' \neq y} \mathbb{E}_\eta[z_\theta^{y'}(x+\eta)] \tag{15}$$

*Then $\tilde{g}_\theta$ is provably robust at x, with the certified radius given by*

$$CR(\tilde{g}_\theta; x, y) = \frac{\sigma}{2}[\Phi^{-1}(\mathbb{E}_\eta[z_\theta^y(x+\eta)]) - \Phi^{-1}(\max_{y' \neq y} \mathbb{E}_\eta[z_\theta^{y'}(x+\eta)])] \tag{16}$$

*where $\Phi$ is the c.d.f. of the standard Gaussian distribution.*

We notice that in Salman et al. (2019) (see its Appendix B), a similar technique was introduced to overcome the non-differentiability in creating adversarial examples to a smoothed classifier. Different from their work, our method uses Soft-RS to obtain a certified radius that is differentiable in practice. The certified radius given by soft randomized smoothing meets condition (C2) in the algorithmic design. Even if we use Monte Carlo sampling to estimate the expectation, (16) is still sub-differentiable with respect to $\theta$ as long as $z_\theta$ is sub-differentiable with respect to $\theta$.

**Connection between Soft-RS and Hard-RS**  We highlight two main properties of Soft-RS. Firstly, it is a differentiable approximation of the original Hard-RS. To see this, note that when $\beta \to \infty$, $z_\theta^y(x; \beta) \xrightarrow{a.e.} \mathbf{1}_{\{y=\arg\max_c u_\theta^c(x)\}}$, so $\tilde{g}_\theta$ converges to $g_\theta$ almost everywhere. Consequently, the Soft-RS certified radius (16) converges to the Hard-RS certified radius (8) almost everywhere as $\beta$ goes to infinity. Secondly, Soft-RS itself provides an alternative way to get a provable robustness guarantee. In Appendix A, we will provide Soft-RS certification procedures that certify $\tilde{g}_\theta$ with the Hoeffding bound or the empirical Bernstein bound.

## 4.4 NUMERICAL STABILITY (FOR CONDITION C3)

In this section, we will address the numerical stability condition (C3). While Soft-RS does provide us with a differentiable certified radius (16) which we could maximize with first-order optimization methods, directly optimizing (16) suffers from exploding gradients. The problem stems from the inverse cumulative density function $\Phi^{-1}(x)$, whose derivative is huge when $x$ is close to 0 or 1.

Fortunately, by minimizing the robustness loss (13) instead, we can maximize the robust radius free from exploding gradients. The hinge loss restricts that samples with non-zero robustness loss must satisfy $0 < CR(\tilde{g}_\theta; x, y) < \epsilon + \tilde{\epsilon}$, which is equivalent to $0 < \xi_\theta(x, y) < \gamma$ where $\xi_\theta(x, y) = \Phi^{-1}(\mathbb{E}_\eta[z_\theta^y(x+\eta)]) - \Phi^{-1}(\max_{y' \neq y} \mathbb{E}_\eta[z_\theta^{y'}(x+\eta)])$ and $\gamma = \frac{2(\epsilon+\tilde{\epsilon})}{\sigma}$. Under this restriction, the derivative of $\Phi^{-1}$ is always bounded as shown in the following proposition. The proof can be found in Appendix B.

**Proposition 1.** *Given any $p_1, p_2, ...p_K$ satisfies $p_1 \geq p_2 \geq ... \geq p_K \geq 0$ and $p_1+p_2+...+p_K = 1$, let $\gamma = \frac{2(\epsilon+\tilde{\epsilon})}{\sigma}$, the derivative of $\min\{[\Phi^{-1}(p_1) - \Phi^{-1}(p_2)], \gamma\}$ with respect to $p_1$ and $p_2$ is bounded.*

### 4.5 COMPLETE IMPLEMENTATION

We are now ready to present the complete MACER algorithm. Expectations over Gaussian samples are approximated with Monte Carlo sampling. Let $\eta_1, \cdots, \eta_k$ be $k$ *i.i.d.* samples from $\mathcal{N}(0, \sigma^2 \boldsymbol{I})$. The final objective function is

$$
\begin{aligned}
l(\tilde{g}_\theta; x, y) &= l_C(\tilde{g}_\theta; x, y) + l_R(\tilde{g}_\theta; x, y) \\
&= -\log \hat{z}_\theta^y(x) + \lambda \cdot \max\{\epsilon + \tilde{\epsilon} - CR(\tilde{g}_\theta; x, y), 0\} \cdot \mathbf{1}_{\{\tilde{g}_\theta(x)=y\}} \\
&= -\log \hat{z}_\theta^y(x) + \frac{\lambda \sigma}{2} \max\{\gamma - \hat{\xi}_\theta(x, y), 0\} \cdot \mathbf{1}_{\{\tilde{g}_\theta(x)=y\}}
\end{aligned}
\tag{17}
$$

where $\hat{z}_\theta(x) = \frac{1}{k} \sum_{j=1}^k z_\theta(x + \eta_j)$ is the empirical expectation of $z_\theta(x + \eta)$ and $\hat{\xi}_\theta(x, y) = \Phi^{-1}(\hat{z}_\theta^y(x)) - \Phi^{-1}(\max_{y' \neq y} \hat{z}_\theta^{y'}(x))$. During training we minimize $\mathbb{E}_{(x,y) \sim \hat{p}_{\text{data}}} l(\tilde{g}_\theta; x, y)$. Detailed implementation is described in Algorithm 1. To simplify the implementation, we choose $\gamma$ to be a hyperparameter instead of $\tilde{\epsilon}$. The inverse temperature of softmax $\beta$ is also a hyperparameter.

---

**Algorithm 1** MACER: robust training via MAximizing CErtified Radius

---

1: **Input:** Training set $\hat{p}_{\text{data}}$, noise level $\sigma$, number of Gaussian samples $k$, trade-off factor $\lambda$, hinge factor $\gamma$, inverse temperature $\beta$, model parameters $\theta$
2: **for** each iteration **do**
3:     Sample a minibatch $(x_1, y_1), \cdots, (x_n, y_n) \sim \hat{p}_{\text{data}}$
4:     For each $x_i$, sample $k$ *i.i.d.* Gaussian samples $x_{i1}, \cdots, x_{ik} \sim \mathcal{N}(x, \sigma^2 \boldsymbol{I})$
5:     Compute the empirical expectations: $\hat{z}_\theta(x_i) \leftarrow \sum_{j=1}^k z_\theta(x_{ij})/k$ for $i = 1, \cdots, n$
6:     Compute $\mathbb{G}_\theta = \{(x_i, y_i) : \tilde{g}_\theta(x_i) = y_i\}$: $(x_i, y_i) \in \mathbb{G}_\theta \Leftrightarrow y_i = \arg\max_{c \in \mathcal{Y}} \hat{z}_\theta^c(x_i)$
7:     For each $(x_i, y_i) \in \mathbb{G}_\theta$, compute $\hat{y}_i$: $\hat{y}_i \leftarrow \arg\max_{c \in \mathcal{Y} \setminus \{y_i\}} \hat{z}_\theta^c(x_i)$
8:     For each $(x_i, y_i) \in \mathbb{G}_\theta$, compute $\hat{\xi}_\theta(x_i, y_i)$: $\hat{\xi}_\theta(x_i, y_i) \leftarrow \Phi^{-1}(\hat{z}_\theta^{y_i}(x_i)) - \Phi^{-1}(\hat{z}_\theta^{\hat{y}_i}(x_i))$
9:     Update $\theta$ with one step of any first-order optimization method to minimize

$$
-\frac{1}{n} \sum_{i=1}^n \log \hat{z}_\theta^{y_i}(x_i) + \frac{\lambda \sigma}{2n} \sum_{(x_i, y_i) \in \mathbb{G}_\theta} \max\{\gamma - \hat{\xi}_\theta(x_i, y_i), 0\}
$$

10: **end for**

---

**Compare to adversarial training**    Adversarial training defines the problem as a mini-max game and solves it by optimizing the inner loop (attack generation) and the outer loop (model update) iteratively. In our method, we only have a single loop (model update). As a result, our proposed algorithm can run much faster than adversarial training because it does not require additional back propagations to generate adversarial examples.

**Compare to previous work**    The overall objective function of our method, a linear combination of a classification loss and a robustness loss, is similar to those of adversarial logit pairing (ALP) (Kannan et al., 2018) and TRADES (Zhang et al., 2019b). In MACER, the $\lambda$ in the objective function (17) can also be viewed as a trade-off factor between accuracy and robustness. However, the robustness term of MACER does not depend on a particular adversarial example $x'$, which makes it substantially different from ALP and TRADES.

## 5 EXPERIMENTS

In this section, we empirically evaluate our proposed MACER algorithm on a wide range of tasks. We also study the influence of different hyperparameters in MACER on the final model performance.

### 5.1 SETUP

To fairly compare with previous works, we follow Cohen et al. (2019) and Salman et al. (2019) to use LeNet for MNIST, ResNet-110 for Cifar-10 and SVHN, and ResNet-50 for ImageNet.

**MACER Training**  For Cifar-10, MNIST and SVHN, we train the models for 440 epochs using our proposed algorithm. The learning rate is initialized to be 0.01, and is decayed by 0.1 at the 200[th]/400[th] epoch. For all the models, we use $k = 16$, $\gamma = 8.0$ and $\beta = 16.0$. The value of $\lambda$ trades off the accuracy and robustness and we find that different $\lambda$ leads to different robust accuracy when the model is injected by different levels ($\sigma$) of noise. We find setting $\lambda = 12.0$ for $\sigma = 0.25$ and $\lambda = 4.0$ for $\sigma = 0.50$ works best. For ImageNet, we train the models for 120 epochs. The initial learning rate is set to be 0.1 and is decayed by 0.1 at the 30[th]/60[th]/90[th] epoch. For all models on ImageNet, we use $k = 2$, $\gamma = 8.0$ and $\beta = 16.0$. More details can be found in Appendix C.

**Baselines**  We compare the performance of MACER with two previous works. The first work (Cohen et al., 2019) trains smoothed networks by simply minimizing cross-entropy loss. The second one (Salman et al., 2019) uses adversarial training on smoothed networks to improve the robustness. For both baselines, we use checkpoints provided by the authors and report their original numbers whenever available. In addition, we run Cohen et al. (2019)'s method on all tasks as it is a speical case of MACER by setting $k = 1$ and $\lambda = 0$.

**Certification**  Following previous works, we report the *approximated certified test set accuracy*, which is the fraction of the test set that can be certified to be robust at radius $r$. However, the approximated certified test set accuracy is a function of the radius $r$. It is hard to compare two models unless one is uniformly better than the other for all $r$. Hence, we also use the *average certified radius* (ACR) as a metric: for each test data $(x, y)$ and model $g$, we can estimate the certified radius $CR(g; x, y)$. The average certified radius is defined as $\frac{1}{|\mathcal{S}_{test}|} \sum_{(x,y) \in \mathcal{S}_{test}} CR(g; x, y)$ where $\mathcal{S}_{test}$ is the test set. To estimate the certified radius for data points, we use the source code provided by Cohen et al. (2019).

## 5.2 RESULTS

We report the results on Cifar-10 and ImageNet in the main body of the paper. Results on MNIST and SVHN can be found in Appendix C.2.

Table 1: Approximated certified test accuracy and ACR on Cifar-10: Each column is an $l_2$ radius.

| $\sigma$ | Model | 0.00 | 0.25 | 0.50 | 0.75 | 1.00 | 1.25 | 1.50 | 1.75 | 2.00 | 2.25 | ACR |
|---|---|---|---|---|---|---|---|---|---|---|---|---|
| | Cohen-0.25 | 0.75 | 0.60 | 0.43 | 0.26 | 0 | 0 | 0 | 0 | 0 | 0 | 0.416 |
| 0.25 | Salman-0.25 | 0.74 | 0.67 | 0.57 | 0.47 | 0 | 0 | 0 | 0 | 0 | 0 | 0.538 |
| | MACER-0.25 | 0.81 | 0.71 | 0.59 | 0.43 | 0 | 0 | 0 | 0 | 0 | 0 | **0.556** |
| | Cohen-0.50 | 0.65 | 0.54 | 0.41 | 0.32 | 0.23 | 0.15 | 0.09 | 0.04 | 0 | 0 | 0.491 |
| 0.50 | Salman-0.50 | 0.50 | 0.46 | 0.44 | 0.40 | 0.38 | 0.33 | 0.29 | 0.23 | 0 | 0 | 0.709 |
| | MACER-0.50 | 0.66 | 0.60 | 0.53 | 0.46 | 0.38 | 0.29 | 0.19 | 0.12 | 0 | 0 | **0.726** |
| | Cohen-1.00 | 0.47 | 0.39 | 0.34 | 0.28 | 0.21 | 0.17 | 0.14 | 0.08 | 0.05 | 0.03 | 0.458 |
| 1.00 | Salman-1.00 | 0.45 | 0.41 | 0.38 | 0.35 | 0.32 | 0.28 | 0.25 | 0.22 | 0.19 | 0.17 | 0.787 |
| | MACER-1.00 | 0.45 | 0.41 | 0.38 | 0.35 | 0.32 | 0.29 | 0.25 | 0.22 | 0.18 | 0.16 | **0.792** |

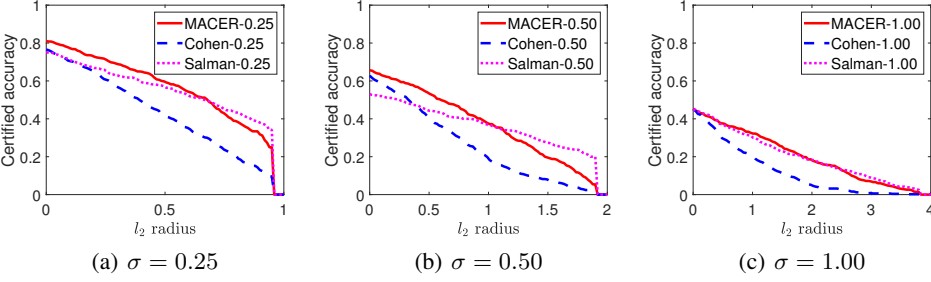

(a) $\sigma = 0.25$       (b) $\sigma = 0.50$       (c) $\sigma = 1.00$

Figure 1: Radius-accuracy curves of different Cifar-10 models.

Table 2: Approximated certified test accuracy and ACR on ImageNet: Each column is an $l_2$ radius.

| $\sigma$ | Model | 0.0 | 0.5 | 1.0 | 1.5 | 2.0 | 2.5 | 3.0 | ACR |
|---|---|---|---|---|---|---|---|---|---|
| | Cohen-0.25 | 0.67 | 0.49 | 0 | 0 | 0 | 0 | 0 | 0.470 |
| 0.25 | Salman-0.25 | 0.65 | 0.56 | 0 | 0 | 0 | 0 | 0 | 0.528 |
| | MACER-0.25 | 0.68 | 0.57 | 0 | 0 | 0 | 0 | 0 | **0.544** |
| | Cohen-0.50 | 0.57 | 0.46 | 0.37 | 0.29 | 0 | 0 | 0 | 0.720 |
| 0.50 | Salman-0.50 | 0.54 | 0.49 | 0.43 | 0.37 | 0 | 0 | 0 | 0.815 |
| | MACER-0.50 | 0.64 | 0.53 | 0.43 | 0.31 | 0 | 0 | 0 | **0.831** |
| | Cohen-1.00 | 0.44 | 0.38 | 0.33 | 0.26 | 0.19 | 0.15 | 0.12 | 0.863 |
| 1.00 | Salman-1.00 | 0.40 | 0.37 | 0.34 | 0.30 | 0.27 | 0.25 | 0.20 | 1.003 |
| | MACER-1.00 | 0.48 | 0.43 | 0.36 | 0.30 | 0.25 | 0.18 | 0.14 | **1.008** |

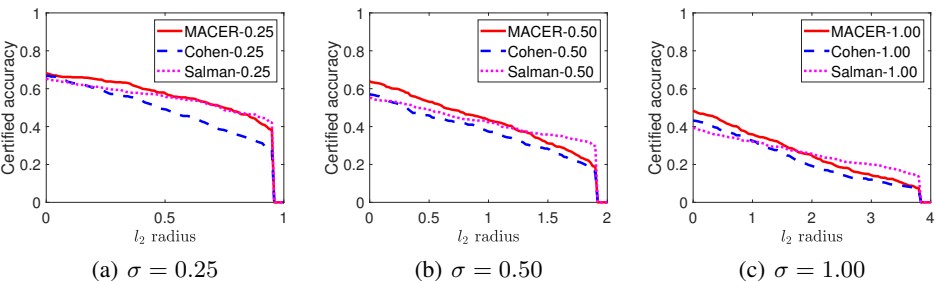

(a) $\sigma = 0.25$       (b) $\sigma = 0.50$       (c) $\sigma = 1.00$

Figure 2: Radius-accuracy curves of different ImageNet models.

**Performance** The performance of different models on Cifar-10 are reported in Table 1, and in Figure 1 we display the radius-accuracy curves. Note that the area under a radius-accuracy curve is equal to the ACR of the model. First, the plots show that our proposed method consistently achieves significantly higher approximated certified test set accuracy than Cohen et al. (2019). This shows that robust training via maximizing the certified radius is more effective than simply minimizing the cross entropy classification loss. Second, the performance of our model is different from that of Salman et al. (2019) for different $r$. For example, for $\sigma = 0.25$, our model achieves higher accuracy than Salman et al. (2019)'s model when $r = 0/0.25/0.5$, but the performance of ours is worse when $r = 0.75$. For the average certified radius, our models are better than Salman et al. (2019)'s models[2] in all settings. For example, when $\sigma = 0.25/0.50$, the ACR of our model is about 3% larger than that of Salman et al. (2019)'s. The gain of our model is relatively smaller when $\sigma = 1.0$. This is because $\sigma = 1.0$ is a very large noise level (Cohen et al., 2019) and both models perform poorly. The ImageNet results are displayed in Table 2 and Figure 2, and the observation is similar. All experimental results show that our proposed algorithm is more effective than previous ones.

**Training speed** Since MACER does not require adversarial attack during training, it runs much faster to learn a robust model. Empirically, we compare MACER with Salman et al. (2019) on the average training time per epoch and the total training hours, and list the statistics in Table 3. For a fair comparison, we use the codes[34] provided by the original authors and run all algorithms on the same machine. For Cifar-10 we use one NVIDIA P100 GPU and for ImageNet we use four NVIDIA P100 GPUs. According to our experiments, on ImageNet, MACER achieves ACR=0.544 in 117.90 hours. On the contrary, Salman et al. (2019) only achieves ACR=0.528 but uses 193.10 hours, which clearly shows that our method is much more efficient.

One might question whether the higher performance of MACER comes from the fact that we train for more epochs than previous methods. In Section C.3 we also run MACER for 150 epochs and compare it with the models in Table 3. The results show that when run for only 150 epochs, MACER still achieves a performance comparable with SmoothAdv, and is 4 times faster at the same time.

---

[2]Salman et al. (2019) releases hundreds of models, and we select the model with the largest average certified radius for each $\sigma$ as our baseline.

[3]`https://github.com/locuslab/smoothing`

[4]`https://github.com/Hadisalman/smoothing-adversarial`

Table 3: Training time and performance of $\sigma = 0.25$ models.

| Dataset | Model | sec/epoch | Epochs | Total hrs | ACR |
|---|---|---|---|---|---|
| Cifar-10 | Cohen-0.25 (Cohen et al., 2019) | 31.4 | 150 | 1.31 | 0.416 |
| | Salman-0.25 (Salman et al., 2019) | 1990.1 | 150 | 82.92 | 0.538 |
| | MACER-0.25 (ours) | 504.0 | 440 | 61.60 | 0.556 |
| ImageNet | Cohen-0.25 (Cohen et al., 2019) | 2154.5 | 90 | 53.86 | 0.470 |
| | Salman-0.25 (Salman et al., 2019) | 7723.8 | 90 | 193.10 | 0.528 |
| | MACER-0.25 (ours) | 3537.1 | 120 | 117.90 | 0.544 |

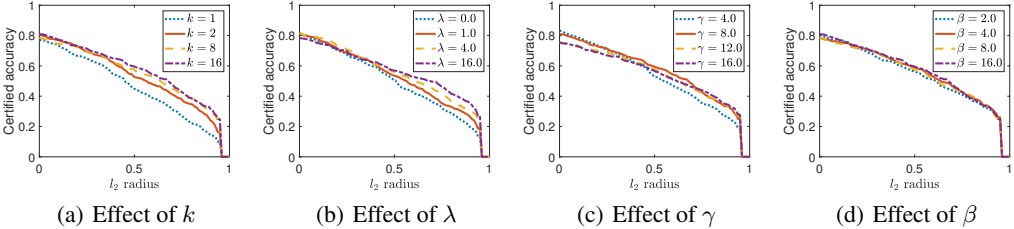

(a) Effect of $k$      (b) Effect of $\lambda$      (c) Effect of $\gamma$      (d) Effect of $\beta$

Figure 3: Effect of hyperparameters on Cifar-10 ($\sigma = 0.25$).

### 5.3 Effect of hyperparameters

In this section, we carefully examine the effect of different hyperparameters in MACER. All experiments are run on Cifar-10 with $\sigma = 0.25$ or $0.50$. The results for $\sigma = 0.25$ are shown in Figure 3. All details can be found in Appendix C.4.

**Effect of $k$** We sample $k$ Gaussian samples for each input to estimate the expectation in (16). We can see from Figure 3(a) that using more Gaussian samples usually leads to better performance. For example, the radius-accuracy curve of $k = 16$ is uniformly above that of $k = 1$.

**Effect of $\lambda$** The radius-accuracy curves in Figure 3(b) demonstrate the trade-off effect of $\lambda$. From the figure, we can see that as $\lambda$ increases, the clean accuracy drops while the certified accuracy at large radii increases.

**Effect of $\gamma$** $\gamma$ is defined as the hyperparameter in the hinge loss. From Figure 3(c) we can see that when $\gamma$ is small, the approximated certified test set accuracy at large radii is small since $\gamma$ "truncates" the large radii. As $\gamma$ increases, the robust accuracy improves. It appears that $\gamma$ also acts as a trade-off between accuracy and robustness, but the effect is not as significant as the effect of $\lambda$.

**Effect of $\beta$** Similar to Salman et al. (2019)'s finding (see its Appendix B), we also observe that using a larger $\beta$ produces better results. While Salman et al. (2019) pointed out that a large $\beta$ may make training unstable, we find that if we only apply a large $\beta$ to the robustness loss, we can maintain training stability and achieve a larger average certified radius as well.

## 6 Conclusion and future work

In this work we propose MACER, an attack-free and scalable robust training method via directly maximizing the certified radius of a smoothed classifier. We discuss the desiderata such an algorithm would have to satisfy, and provide an approach to each of them. According to our extensive experiments, MACER performs better than previous provable $l_2$-defenses and trains faster. Our strong empirical results suggest that adversarial training is not a must for robust training, and defense based on certification is a promising direction for future research. Moreover, several recent papers (Carmon et al., 2019; Zhai et al., 2019; Stanforth et al., 2019) suggest that using unlabeled data helps improve adversarially robust generalization. We will also extend MACER to the semi-supervised setting.

ACKNOWLEDGMENTS

We thank Tianle Cai for helpful discussions and suggestions. This work was done when Runtian Zhai was visiting UCLA under the Top-Notch Undergraduate Program of Peking University school of EECS. Chen Dan and Pradeep Ravikumar acknowledge the support of Rakuten Inc., and NSF via IIS1909816. Huan Zhang and Cho-Jui Hsieh acknowledge the support of NSF via IIS1719097. Liwei Wang acknowledges the support of Beijing Academy of Artificial Intelligence.

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

# A  SOFT RANDOMIZED SMOOTHING

In this section we provide theoretical analysis and certification procedures for Soft-RS.

## A.1  PROOF OF THEOREM 2

Our proof is based on the following lemma:

**Lemma 1.** *For any measurable function $f : \mathcal{X} \to [0, 1]$, define $\hat{f}(x) = \mathbb{E}_{\eta \sim \mathcal{N}(0,\sigma^2 I)} f(x + \eta)$, then $x \mapsto \Phi^{-1}(\hat{f}(x))$ is $1/\sigma$-Lipschitz.*

This lemma is the generalized version of Lemma 2 in Salman et al. (2019).

*Proof of Theorem 2.* Let $y^* = \arg\max_{y' \neq y} \mathbb{E}_\eta[z_\theta^{y'}(x + \eta)]$. For any $c \in \mathcal{Y}$, define $\hat{z}_\theta^c$ as:

$$\hat{z}_\theta^c(x) = \mathbb{E}_{\eta \sim \mathcal{N}(0,\sigma^2 I)}[z_\theta^c(x + \eta)] \tag{18}$$

Because $z_\theta^c : \mathcal{X} \to [0, 1]$, by Lemma 1 we have $x \mapsto \Phi^{-1}(\hat{z}_\theta^c(x))$ is $1/\sigma$-Lipschitz. Thus, $\forall y' \neq y$, for any $\delta$ such that $\|\delta\|_2 \leq \frac{\sigma}{2}[\Phi^{-1}(\mathbb{E}_\eta[z_\theta^y(x + \eta)]) - \Phi^{-1}(\max_{y' \neq y} \mathbb{E}_\eta[z_\theta^{y'}(x + \eta)])]$:

$$\Phi^{-1}(\hat{z}_\theta^y(x + \delta)) \geq \Phi^{-1}(\hat{z}_\theta^y(x)) - \frac{1}{2}[\Phi^{-1}(\hat{z}_\theta^y(x)) - \Phi^{-1}(\hat{z}_\theta^{y^*}(x))]$$

$$\Phi^{-1}(\hat{z}_\theta^{y'}(x + \delta)) \leq \Phi^{-1}(\hat{z}_\theta^{y'}(x)) + \frac{1}{2}[\Phi^{-1}(\hat{z}_\theta^y(x)) - \Phi^{-1}(\hat{z}_\theta^{y^*}(x))] \tag{19}$$

$$\leq \Phi^{-1}(\hat{z}_\theta^{y^*}(x)) + \frac{1}{2}[\Phi^{-1}(\hat{z}_\theta^y(x)) - \Phi^{-1}(\hat{z}_\theta^{y^*}(x))]$$

Therefore, $\Phi^{-1}(\mathbb{E}_\eta z_\theta^y(x + \delta + \eta)) \geq \Phi^{-1}(\mathbb{E}_\eta z_\theta^{y'}(x + \delta + \eta))$. Due to the monotonicity of $\Phi^{-1}$, we have $\mathbb{E}_\eta z_\theta^y(x + \delta + \eta) \geq \mathbb{E}_\eta z_\theta^{y'}(x + \delta + \eta)$, which implies that $\tilde{g}_\theta(x + \delta) = y$. $\square$

## A.2  SOFT-RS CERTIFICATION PROCEDURE

Let $z_A = \mathbb{E}_\eta[z_\theta^y(x + \eta)]$ and $z_B = \max_{y' \neq y} \mathbb{E}_\eta[z_\theta^{y'}(x + \eta)]$. If there exist $\underline{z_A}, \overline{z_B} \in [0, 1]$ such that $P(z_A \geq \underline{z_A} \wedge z_B \leq \overline{z_B}) \geq 1 - \alpha$, then with probability at least $1 - \alpha$, $CR(\tilde{g}_\theta; x, y) \geq \frac{\sigma}{2}[\Phi^{-1}(\underline{z_A}) - \Phi^{-1}(\overline{z_B})]$. Meanwhile, $z_B \leq 1 - z_A$, so we can take $\overline{z_B} = 1 - \underline{z_A}$, and

$$P(CR(\tilde{g}_\theta; x, y) \geq \sigma \Phi^{-1}(\underline{z_A})) \geq 1 - \alpha \tag{20}$$

It reduces to find a confidence lower bound of $z_A$. Here we provide two bounds:

**Hoeffding Bound**   The random variable $z_\theta^y(x + \eta)$ has mean $z_A$, and $z_1^y, \cdots, z_k^y$ are its $k$ observations. Because $z_j^y \in [0, 1]$ for any $j = 1, \cdots, k$, we can use Hoeffding's inequality to obtain a lower confidence bound:

**Lemma 2.** *(Hoeffding's Inequality) Let $X_1, ..., X_k$ be independent random variables bounded by the interval $[0, 1]$. Let $\overline{X} = \frac{1}{k} \sum_{j=1}^k X_j$, then for any $t \geq 0$*

$$P(\overline{X} - \mathbb{E}X \geq t) \leq e^{-2kt^2} \tag{21}$$

Denote $\hat{z}^y = \frac{1}{k} \sum_{j=1}^k z_j^y$. By Hoeffding's inequality we have

$$P(\hat{z}^y - z_A \geq \sqrt{\frac{-\log \alpha}{2k}}) \leq \alpha \tag{22}$$

Hence, a $1 - \alpha$ confidence lower bound $\underline{z_A}$ of $z_A$ is

$$\underline{z_A} = \hat{z}^y - \sqrt{\frac{-\log \alpha}{2k}} \tag{23}$$

**Empirical Bernstein Bound**    Maurer & Pontil (2009) provides us with a tighter bound:

**Theorem 3.** *(Theorem 4 in Maurer & Pontil (2009)) Under the conditions of Lemma 2, with probability at least $1 - \alpha$,*

$$\overline{X} - \mathbb{E}X \leq \sqrt{\frac{2S^2 \log \frac{2}{\alpha}}{k}} + \frac{7 \log \frac{2}{\alpha}}{3(k-1)} \tag{24}$$

*where $S^2$ is the sample variance of $X_1, \cdots, X_k$, i.e.*

$$S^2 = \frac{\sum_{j=1}^k X_j^2 - \frac{(\sum_{j=1}^k X_j)^2}{k}}{k-1} \tag{25}$$

Consequently, a $1 - \alpha$ confidence lower bound $\underline{z_A}$ of $z_A$ is

$$\underline{z_A} = \hat{z}^y - \sqrt{\frac{2S^2 \log \frac{2}{\alpha}}{k}} - \frac{7 \log \frac{2}{\alpha}}{3(k-1)} \tag{26}$$

The full certification procedure with the above two bounds is described in Algorithm 2.

---

**Algorithm 2** Soft randomized smoothing certification

---

1: *# Certify the robustness of $\tilde{g}$ around an input $x$ with Hoeffding bound*
2: **function** CERTIFYHOEFFDING($z, \sigma^2, x, n_0, n, \alpha$)
3:     $\hat{z_0} \leftarrow$ SAMPLEUNDERNOISE($z, x, n_0, \sigma^2$)$[1, :]/n_0$
4:     $\hat{c}_A \leftarrow \arg\max_c \hat{z_0}^c$
5:     $\hat{z}_A \leftarrow$ SAMPLEUNDERNOISE($z, x, n, \sigma^2$)$[1, \hat{c}_A]/n$
6:     $\underline{z_A} \leftarrow \hat{z}_A - \sqrt{\frac{-\log \alpha}{2n}}$
7:     **if** $\underline{z_A} > \frac{1}{2}$ **then return** prediction $\hat{c}_A$ and radius $\sigma\Phi^{-1}(\underline{z_A})$
8:     **else return** ABSTAIN
9: **end function**

10: *# Certify with empirical Bernstein bound*
11: **function** CERTIFYBERNSTEIN($z, \sigma^2, x, n_0, n, \alpha$)
12:     $\hat{z_0} \leftarrow$ SAMPLEUNDERNOISE($z, x, n_0, \sigma^2$)$[1, :]/n_0$
13:     $\hat{c}_A \leftarrow \arg\max_c \hat{z_0}^c$
14:     $A \leftarrow$ SAMPLEUNDERNOISE($z, x, n, \sigma^2$)
15:     $\hat{z}_A \leftarrow A[1, \hat{c}_A]/n, S_A^2 \leftarrow \frac{A[2, \hat{c}_A] - A[1, \hat{c}_A]^2/n}{n-1}$
16:     $\underline{z_A} \leftarrow \hat{z}_A - \sqrt{\frac{2S_A^2 \log(2/\alpha)}{n}} - \frac{7 \log(2/\alpha)}{3(n-1)}$
17:     **if** $\underline{z_A} > \frac{1}{2}$ **then return** prediction $\hat{c}_A$ and radius $\sigma\Phi^{-1}(\underline{z_A})$
18:     **else return** ABSTAIN
19: **end function**

20: *# Helper function: draw num samples from $z(x + \eta)$ and return the $1^{st}$ and $2^{nd}$ sample moment*
21: **function** SAMPLEUNDERNOISE($z, x, \text{num}, \sigma^2$)
22:     Initialize a $2 \times K$ matrix $A \leftarrow (0, \cdots, 0; 0, \cdots, 0)$
23:     **for** $j = 1$ **to** num **do**
24:         Sample noise $\eta_j \sim \mathcal{N}(0, \sigma^2 \boldsymbol{I})$
25:         Compute: $z_j = z(x + \eta_j)$
26:         Increment: $A[1, :] \leftarrow A[1, :] + z_j, A[2, :] \leftarrow A[2, :] + z_j^2$
27:     **end for**
28:     **return** $A$
29: **end function**

---

### A.3    COMPARING SOFT-RS WITH HARD-RS

We use Soft-RS during training and use Hard-RS during certification. In this section, we empirically compare these two certification methods. We certify the nine models in Table 4. For each model,

we certify with both Hard-RS and Soft-RS. For Hard-RS, we use Clopper-Pearson bound and for Soft-RS, we use the empirical Bernstein bound with different choices of $\beta$. The results are displayed in Figure 4. The results show that Hard-RS consistently gives a larger lower bound of robust radius than Soft-RS. We also observe that there is a gap between Soft-RS and Hard-RS when $\beta \to \infty$, which implies that the empirical Bernstein bound, though tighter than the Hoeffding bound, is still looser than the Clopper-Pearson bound.

## B   PROOF OF PROPOSITION 1

*Proof of Proposition 1.* We only need to consider the case when $\Phi^{-1}(p_1) - \Phi^{-1}(p_2) \leq \gamma$ since the derivative is zero when $\Phi^{-1}(p_1) - \Phi^{-1}(p_2) > \gamma$. Obviously, $p_2 \leq 0.5$ and thus $\Phi^{-1}(p_2) \leq 0$. So $\Phi^{-1}(p_1) \leq \gamma$.

Define $p^* \in (0,1)$ such that $\Phi^{-1}(p^*) = \gamma$. Since $\Phi^{-1}(p)$ is a strictly increasing function of $p$, $p^*$ is unique, and $p_1 \leq p^*$. $\min\{[\Phi^{-1}(p_1) - \Phi^{-1}(p_2)], \gamma\} = \Phi^{-1}(p_1) - \Phi^{-1}(p_2)$. Since $p_1$ is the largest value and $p_1 + p_2 + ... + p_K = 1$, we have $\frac{1}{K} \leq p_1 \leq p^*$. Since $[\Phi^{-1}]'(p)$ is continuous in any closed interval of $(0,1)$, the derivative of $\Phi^{-1}(p_1) - \Phi^{-1}(p_2)$ with respect to $p_1$ is bounded. Similarly, $p_2$ is the largest among $p_2, ...p_K$ and $(K-1)p_2 \geq p_2 + ... + p_K = 1 - p_1 \geq 1 - p^*$. Thus $1 - p^* \geq p_2 \geq \frac{1-p^*}{K-1}$, and the derivative of $\Phi^{-1}(p_1) - \Phi^{-1}(p_2)$ with respect to $p_2$ is bounded. $\square$

## C   SUPPLEMENTARY MATERIAL FOR EXPERIMENTS

### C.1   COMPARED MODELS

In this section we list all compared models in the main body of this paper. Cifar-10 models are listed in Table 4, and ImageNet models are listed in Table 5.

Table 4: Models for comparison on Cifar-10.

| $\sigma$ | Model | Description |
|---|---|---|
| Cohen-{0.25,0.50,1.00} | | Cohen et al. (2019)'s models |
| 0.25 | Salman-0.25 | 8-sample 10-step SmoothAdv$_{PGD}$ with $\epsilon = 1.00$ |
| | MACER-0.25 | MACER with $k = 16$, $\lambda = 12.0$, $\beta = 16.0$ and $\gamma = 8.0$ |
| 0.50 | Salman-0.50 | 2-sample 10-step SmoothAdv$_{PGD}$ with $\epsilon = 2.00$ |
| | MACER-0.50 | MACER with $k = 16$, $\lambda = 4.0$, $\beta = 16.0$ and $\gamma = 8.0$ |
| 1.00 | Salman-1.00 | 2-sample 10-step SmoothAdv$_{PGD}$ with $\epsilon = 2.00$ |
| | MACER-1.00 | MACER with $k = 16$, dynamic $\lambda^5$, $\beta = 16.0$ and $\gamma = 8.0$ |

Table 5: Models for comparison on ImageNet.

| $\sigma$ | Model | Description |
|---|---|---|
| Cohen-{0.25,0.50,1.00} | | Cohen et al. (2019)'s models |
| 0.25 | Salman-0.25 | 1-sample 2-step SmoothAdv$_{DDN}$ with $\epsilon = 1.0$ |
| | MACER-0.25 | MACER with $k = 2$, $\lambda = 6.0$, $\beta = 16.0$ and $\gamma = 8.0$ |
| 0.50 | Salman-0.50 | 1-sample 1-step SmoothAdv$_{PGD}$ with $\epsilon = 1.0$ |
| | MACER-0.50 | MACER with $k = 2$, $\lambda = 3.0$, $\beta = 16.0$ and $\gamma = 8.0$ |
| 1.00 | Salman-1.00 | 1-sample 1-step SmoothAdv$_{PGD}$ with $\epsilon = 2.0$ |
| | MACER-1.00 | MACER with $k = 2$, $\lambda = 3.0$, $\beta = 16.0$ and $\gamma = 8.0$ |

### C.2   RESULTS ON MNIST AND SVHN

Here we present experimental results on MNIST and SVHN. For comparison we also report the results produced by Cohen et al. (2019)'s method.

---

[5]We first train with $\lambda = 0.0$, and then change to $\lambda = 12.0$ after learning rate decay.

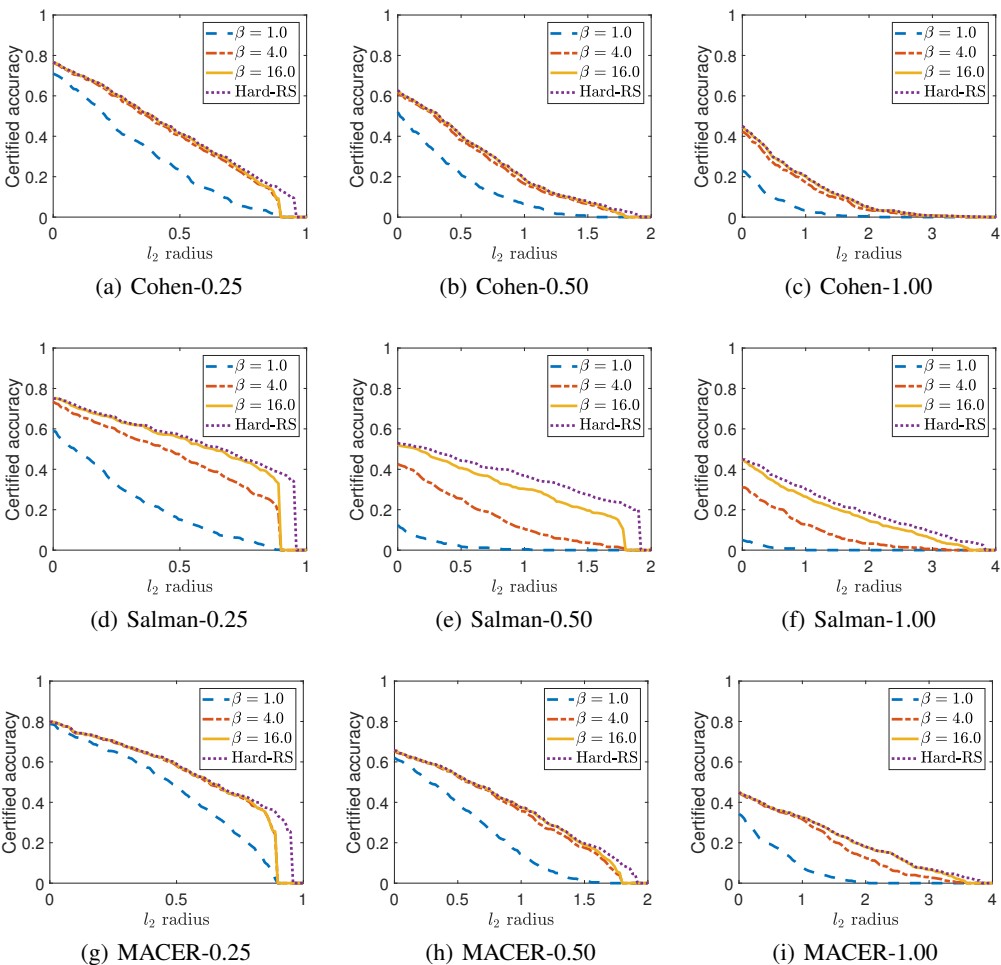

Figure 4: Comparing Soft-RS with Hard-RS on Cifar-10. The three columns correspond to $\sigma = 0.25, 0.50, 1.00$ respectively.

### C.2.1    MNIST

The results are reported in Table 6. For all $\sigma$, we use $k = 16$, $\lambda = 16.0$, $\gamma = 8.0$ and $\beta = 16.0$.

Table 6: Approximated certified test accuracy and ACR on MNIST: Each column is an $l_2$ radius.

| $\sigma$ | Method | 0.00 | 0.25 | 0.50 | 0.75 | 1.00 | 1.25 | 1.50 | 1.75 | 2.00 | 2.25 | ACR |
|------|--------|------|------|------|------|------|------|------|------|------|------|------|
| 0.25 | Cohen | 0.99 | 0.97 | 0.94 | 0.89 | 0 | 0 | 0 | 0 | 0 | 0 | 0.887 |
|      | MACER | 0.99 | 0.99 | 0.97 | 0.95 | 0 | 0 | 0 | 0 | 0 | 0 | **0.918** |
| 0.50 | Cohen | 0.99 | 0.97 | 0.94 | 0.91 | 0.84 | 0.75 | 0.57 | 0.33 | 0 | 0 | 1.453 |
|      | MACER | 0.99 | 0.98 | 0.96 | 0.94 | 0.90 | 0.83 | 0.73 | 0.50 | 0 | 0 | **1.583** |
| 1.00 | Cohen | 0.95 | 0.92 | 0.87 | 0.81 | 0.72 | 0.61 | 0.50 | 0.34 | 0.20 | 0.10 | 1.417 |
|      | MACER | 0.89 | 0.85 | 0.79 | 0.75 | 0.69 | 0.61 | 0.54 | 0.45 | 0.36 | 0.28 | **1.520** |

### C.2.2    SVHN

The results are reported in Table 7. We use $k = 16$, $\lambda = 12.0$, $\gamma = 8.0$ and $\beta = 16.0$.

Table 7: Approximated certified test accuracy and ACR on SVHN: Each column is an $l_2$ radius.

| $\sigma$ | Method | 0.00 | 0.25 | 0.50 | 0.75 | 1.00 | 1.25 | 1.50 | 1.75 | 2.00 | 2.25 | ACR |
|------|--------|------|------|------|------|------|------|------|------|------|------|------|
| 0.25 | Cohen | 0.90 | 0.70 | 0.44 | 0.26 | 0 | 0 | 0 | 0 | 0 | 0 | 0.469 |
|      | MACER | 0.86 | 0.72 | 0.56 | 0.39 | 0 | 0 | 0 | 0 | 0 | 0 | **0.540** |
| 0.50 | Cohen | 0.67 | 0.48 | 0.37 | 0.24 | 0.14 | 0.08 | 0.06 | 0.03 | 0 | 0 | 0.434 |
|      | MACER | 0.61 | 0.53 | 0.44 | 0.35 | 0.24 | 0.15 | 0.09 | 0.04 | 0 | 0 | **0.538** |

### C.3    MACER TRAINING FOR 150 EPOCHS

In Table 8 we report the performance and training time of MACER on Cifar-10 when it is only run for 150 epochs, and compare with SmoothAdv (Salman et al., 2019) and MACER (440 epochs). The learning rate is decayed by 0.1 at epochs 60 and 120. All other hyperparameters are kept the same as in Table 4.

Table 8: Approximated certified test accuracy and ACR on Cifar-10: Each column is an $l_2$ radius.

| $\sigma$ | Method | 0.00 | 0.25 | 0.50 | 0.75 | 1.00 | 1.25 | 1.50 | 1.75 | ACR | Epochs | Total hrs |
|------|--------|------|------|------|------|------|------|------|------|------|------|------|
|      | SmoothAdv | 0.74 | 0.67 | 0.57 | 0.47 | 0 | 0 | 0 | 0 | 0.538 | 150 | 82.92 |
| 0.25 | MACER | 0.76 | 0.67 | 0.57 | 0.42 | 0 | 0 | 0 | 0 | 0.531 | 150 | 21.00 |
|      | MACER | 0.81 | 0.71 | 0.59 | 0.43 | 0 | 0 | 0 | 0 | 0.556 | 440 | 61.60 |
|      | SmoothAdv | 0.50 | 0.46 | 0.44 | 0.40 | 0.38 | 0.33 | 0.29 | 0.23 | 0.709 | 150 | 82.92 |
| 0.50 | MACER | 0.62 | 0.57 | 0.50 | 0.44 | 0.38 | 0.29 | 0.21 | 0.13 | 0.712 | 150 | 21.00 |
|      | MACER | 0.66 | 0.60 | 0.53 | 0.46 | 0.38 | 0.29 | 0.19 | 0.12 | 0.726 | 440 | 61.60 |

### C.4    EFFECT OF HYPERPARAMETERS

All experiments are run on Cifar-10 with $\sigma = 0.25$ or $0.50$. See Table 9 for detailed experimental settings. Results are reported in Tables 10-13.

Table 9: Experimental setting for examining the effect of hyperparameters.

| Experiment | $k$ | $\lambda$ | $\gamma$ | $\beta$ |
|---|---|---|---|---|
| Effect of $k$ | 1/2/4/8/16 | 12.0 | 8.0 | 16.0 |
| Effect of $\lambda$ | 16 | 0.0/1.0/2.0/4.0/8.0/16.0 | 8.0 | 16.0 |
| Effect of $\gamma$ | 16 | 12.0 | 2.0/4.0/6.0/8.0/10.0/12.0/14.0/16.0 | 16.0 |
| Effect of $\beta$ | 16 | 12.0 | 8.0 | 1.0/2.0/4.0/8.0/16.0/32.0/64.0 |

Table 10: Effect of $k$: Approximated certified test accuracy and ACR on Cifar-10.

| $\sigma$ | $k$ | 0.00 | 0.25 | 0.50 | 0.75 | 1.00 | 1.25 | 1.50 | 1.75 | 2.00 | 2.25 | ACR |
|---|---|---|---|---|---|---|---|---|---|---|---|---|
| | 1 | 0.77 | 0.65 | 0.45 | 0.27 | 0 | 0 | 0 | 0 | 0 | 0 | 0.448 |
| | 2 | 0.81 | 0.69 | 0.53 | 0.38 | 0 | 0 | 0 | 0 | 0 | 0 | 0.519 |
| 0.25 | 4 | 0.79 | 0.69 | 0.55 | 0.38 | 0 | 0 | 0 | 0 | 0 | 0 | 0.517 |
| | 8 | 0.78 | 0.69 | 0.57 | 0.41 | 0 | 0 | 0 | 0 | 0 | 0 | 0.538 |
| | 16 | 0.81 | 0.71 | 0.59 | 0.43 | 0 | 0 | 0 | 0 | 0 | 0 | 0.556 |
| | 1 | 0.36 | 0.31 | 0.26 | 0.19 | 0.12 | 0.09 | 0.05 | 0.02 | 0 | 0 | 0.306 |
| | 2 | 0.60 | 0.53 | 0.47 | 0.37 | 0.28 | 0.19 | 0.13 | 0.08 | 0 | 0 | 0.589 |
| 0.50 | 4 | 0.60 | 0.55 | 0.50 | 0.42 | 0.34 | 0.26 | 0.18 | 0.11 | 0 | 0 | 0.665 |
| | 8 | 0.61 | 0.56 | 0.50 | 0.43 | 0.37 | 0.28 | 0.22 | 0.14 | 0 | 0 | 0.699 |
| | 16 | 0.60 | 0.56 | 0.50 | 0.46 | 0.38 | 0.29 | 0.23 | 0.14 | 0 | 0 | 0.712 |

Table 11: Effect of $\lambda$: Approximated certified test accuracy and ACR on Cifar-10.

| $\sigma$ | $\lambda$ | 0.00 | 0.25 | 0.50 | 0.75 | 1.00 | 1.25 | 1.50 | 1.75 | 2.00 | 2.25 | ACR |
|---|---|---|---|---|---|---|---|---|---|---|---|---|
| | 0.0 | 0.82 | 0.67 | 0.51 | 0.32 | 0 | 0 | 0 | 0 | 0 | 0 | 0.488 |
| | 1.0 | 0.81 | 0.68 | 0.54 | 0.35 | 0 | 0 | 0 | 0 | 0 | 0 | 0.516 |
| 0.25 | 2.0 | 0.81 | 0.71 | 0.54 | 0.36 | 0 | 0 | 0 | 0 | 0 | 0 | 0.523 |
| | 4.0 | 0.82 | 0.71 | 0.56 | 0.41 | 0 | 0 | 0 | 0 | 0 | 0 | 0.540 |
| | 8.0 | 0.80 | 0.70 | 0.59 | 0.43 | 0 | 0 | 0 | 0 | 0 | 0 | 0.550 |
| | 16.0 | 0.78 | 0.69 | 0.57 | 0.46 | 0 | 0 | 0 | 0 | 0 | 0 | 0.550 |
| | 0.0 | 0.69 | 0.56 | 0.44 | 0.31 | 0.21 | 0.13 | 0.06 | 0.02 | 0 | 0 | 0.515 |
| | 1.0 | 0.68 | 0.59 | 0.52 | 0.43 | 0.33 | 0.23 | 0.15 | 0.07 | 0 | 0 | 0.662 |
| 0.50 | 2.0 | 0.70 | 0.61 | 0.53 | 0.43 | 0.36 | 0.27 | 0.18 | 0.09 | 0 | 0 | 0.709 |
| | 4.0 | 0.66 | 0.60 | 0.53 | 0.46 | 0.37 | 0.29 | 0.19 | 0.12 | 0 | 0 | 0.726 |
| | 8.0 | 0.62 | 0.56 | 0.50 | 0.45 | 0.37 | 0.29 | 0.21 | 0.13 | 0 | 0 | 0.700 |
| | 16.0 | 0.60 | 0.56 | 0.51 | 0.45 | 0.36 | 0.30 | 0.21 | 0.14 | 0 | 0 | 0.711 |

Table 12: Effect of $\gamma$: Approximated certified test accuracy and ACR on Cifar-10.

| $\sigma$ | $\gamma$ | 0.00 | 0.25 | 0.50 | 0.75 | 1.00 | 1.25 | 1.50 | 1.75 | 2.00 | 2.25 | ACR |
|---|---|---|---|---|---|---|---|---|---|---|---|---|
| | 2.0 | 0.82 | 0.67 | 0.47 | 0.26 | 0 | 0 | 0 | 0 | 0 | 0 | 0.455 |
| | 4.0 | 0.83 | 0.70 | 0.53 | 0.35 | 0 | 0 | 0 | 0 | 0 | 0 | 0.521 |
| | 6.0 | 0.80 | 0.70 | 0.57 | 0.40 | 0 | 0 | 0 | 0 | 0 | 0 | 0.539 |
| 0.25 | 8.0 | 0.81 | 0.71 | 0.59 | 0.43 | 0 | 0 | 0 | 0 | 0 | 0 | 0.556 |
| | 10.0 | 0.78 | 0.69 | 0.57 | 0.44 | 0 | 0 | 0 | 0 | 0 | 0 | 0.543 |
| | 12.0 | 0.76 | 0.69 | 0.58 | 0.42 | 0 | 0 | 0 | 0 | 0 | 0 | 0.534 |
| | 14.0 | 0.76 | 0.68 | 0.55 | 0.44 | 0 | 0 | 0 | 0 | 0 | 0 | 0.536 |
| | 16.0 | 0.75 | 0.67 | 0.56 | 0.44 | 0 | 0 | 0 | 0 | 0 | 0 | 0.534 |
| | 2.0 | 0.70 | 0.61 | 0.49 | 0.33 | 0.23 | 0.13 | 0.06 | 0.03 | 0 | 0 | 0.549 |
| | 4.0 | 0.68 | 0.60 | 0.53 | 0.44 | 0.34 | 0.25 | 0.15 | 0.06 | 0 | 0 | 0.676 |
| | 6.0 | 0.65 | 0.58 | 0.51 | 0.43 | 0.35 | 0.28 | 0.20 | 0.13 | 0 | 0 | 0.706 |
| 0.50 | 8.0 | 0.60 | 0.56 | 0.50 | 0.46 | 0.38 | 0.29 | 0.23 | 0.14 | 0 | 0 | 0.712 |
| | 10.0 | 0.58 | 0.54 | 0.49 | 0.44 | 0.37 | 0.31 | 0.24 | 0.15 | 0 | 0 | 0.707 |
| | 12.0 | 0.56 | 0.51 | 0.48 | 0.43 | 0.38 | 0.34 | 0.26 | 0.16 | 0 | 0 | 0.712 |
| | 14.0 | 0.51 | 0.48 | 0.44 | 0.38 | 0.33 | 0.28 | 0.22 | 0.16 | 0 | 0 | 0.634 |
| | 16.0 | 0.57 | 0.54 | 0.47 | 0.41 | 0.34 | 0.30 | 0.24 | 0.17 | 0 | 0 | 0.695 |

Table 13: Effect of $\beta$: Approximated certified test accuracy and ACR on Cifar-10.

| $\sigma$ | $\beta$ | 0.00 | 0.25 | 0.50 | 0.75 | 1.00 | 1.25 | 1.50 | 1.75 | 2.00 | 2.25 | ACR |
|---|---|---|---|---|---|---|---|---|---|---|---|---|
| | 1.0 | 0.79 | 0.67 | 0.54 | 0.37 | 0 | 0 | 0 | 0 | 0 | 0 | 0.513 |
| | 2.0 | 0.81 | 0.69 | 0.56 | 0.41 | 0 | 0 | 0 | 0 | 0 | 0 | 0.534 |
| | 4.0 | 0.79 | 0.70 | 0.59 | 0.43 | 0 | 0 | 0 | 0 | 0 | 0 | 0.549 |
| 0.25 | 8.0 | 0.79 | 0.71 | 0.57 | 0.43 | 0 | 0 | 0 | 0 | 0 | 0 | 0.541 |
| | 16.0 | 0.81 | 0.71 | 0.59 | 0.43 | 0 | 0 | 0 | 0 | 0 | 0 | 0.556 |
| | 32.0 | 0.79 | 0.70 | 0.58 | 0.43 | 0 | 0 | 0 | 0 | 0 | 0 | 0.549 |
| | 64.0 | 0.74 | 0.65 | 0.54 | 0.42 | 0 | 0 | 0 | 0 | 0 | 0 | 0.517 |
| | 1.0 | 0.67 | 0.59 | 0.52 | 0.43 | 0.35 | 0.26 | 0.19 | 0.10 | 0 | 0 | 0.696 |
| | 2.0 | 0.63 | 0.57 | 0.52 | 0.45 | 0.37 | 0.31 | 0.22 | 0.13 | 0 | 0 | 0.719 |
| | 4.0 | 0.60 | 0.55 | 0.50 | 0.45 | 0.36 | 0.30 | 0.22 | 0.14 | 0 | 0 | 0.703 |
| 0.50 | 8.0 | 0.60 | 0.56 | 0.52 | 0.43 | 0.38 | 0.30 | 0.24 | 0.14 | 0 | 0 | 0.713 |
| | 16.0 | 0.60 | 0.56 | 0.50 | 0.46 | 0.38 | 0.29 | 0.23 | 0.14 | 0 | 0 | 0.712 |
| | 32.0 | 0.59 | 0.55 | 0.49 | 0.44 | 0.36 | 0.30 | 0.23 | 0.14 | 0 | 0 | 0.705 |
| | 64.0 | 0.45 | 0.41 | 0.38 | 0.35 | 0.32 | 0.26 | 0.21 | 0.16 | 0 | 0 | 0.579 |

