# OpenReview forum: "MACER: Attack-free and Scalable Robust Training via Maximizing Certified Radius"
_ICLR.cc/2020/Conference — Accept (Poster)_

### Official Review · AnonReviewer3 · 2019-10-17
**Official Blind Review #3**

**Rating:** 3

**Review:**

The paper presents a method for training a certified robust neural network, based on the certification method of cohen et al.

I think this paper is quiet borderline, as it is a natural extension of cohen et al, but is incremental. The authors use a hinge surrogate loss with smoothing to make the certification differentiable and optimize it, getting results that are on par with Salmans et al.

Detailed remarks:
 - From table 3 it seems like MACER was trained for more then the other models, up to almost x3 times more on cifar. This makes the comparison unfair and puts the results into question.
- It was nice that the authors gave a theoretical guarantee for the soft-RS, but it is not clear if that is needed. You can train with soft-RS and apply the hard-RS at test time which has the standard guarantees
- Fig. 3 needs labels on x&y axis
-  The ablation study isn't really an ablation study, it is more testing the sensitivity of various parameters (which is good in itself). However the gamma results and claims are hard to see from the plot.



**Experience Assessment:**

I have published one or two papers in this area.

**Review Assessment: Checking Correctness Of Derivations And Theory:**

I carefully checked the derivations and theory.

**Review Assessment: Checking Correctness Of Experiments:**

I carefully checked the experiments.

**Review Assessment: Thoroughness In Paper Reading:**

I read the paper thoroughly.

---

> ### Author Response · Authors · 2019-11-09
> **Author Response (Part 1)**
>
> Dear AnonReviewer3,
>
> Thank you for thoroughly reading our paper and carefully checking our theoretical analysis and experiments. We address your concerns as follows:
>
> [Regarding the novelty of the paper]
>
> We respectfully disagree with the comment that our work is incremental and is a natural extension of Cohen et al. We would like to clarify the differences between the two works and state the novelty of our paper below:
>
> (a) Cohen et al. analyzed adversarial robustness via Gaussian smoothing of a given base classifier, and suggested training a base classifier by augmenting data with Gaussian noise, which cannot lead to models with high certified robust accuracy and large certified radii. Indeed, they pose it as an open problem the difficult task of training models that explicitly maximize the *certified radius*: this is precisely what we do. It should also be emphasized that the instantiation of this idea comes with considerable technical challenges: we list these explicitly in Section 4.1, and derive the MACER algorithm step by step to make sure that it satisfies all these necessary desiderata. For example, the hinge loss based surrogate of the robustness error is not heuristically picked, but carefully chosen to solve numerical issues. Please also see the discussion in the next point on the methodological novelty of our paper. As for the performance, the experimental results clearly illustrate that MACER performs uniformly better than Cohen et al. by a large margin on all 4 tasks.
>
> (b) We also respectfully disagree that our work is "on par" with Salman et al. Firstly, our approach is methodologically novel. Most previous papers about (provable) adversarial defenses, including Salman et al., were based on adversarial training, and there is even folklore that adversarial training is a must to train robust models. However, MACER trains without adversarial training and still performs better than all previous provable $l_2$ defenses. To our best knowledge, MACER is the first *attack-free* robust training method that is scalable to datasets as large as ImageNet. Our extensive experiments demonstrate the efficiency and effectiveness of MACER, showing that adversarial training is not a must and invalidating the widespread folklore. Second, on all tasks, our model is better than Salman et al.'s and is trained much faster as we do not use adversarial training. For example, our algorithm runs 34.61% faster and performs 3.35% better than Salman et al.'s best model on CIFAR-10 ($\sigma$ = 0.25).
>
> As stated in (a) and (b), given such significant differences from previous work, rigorous derivation of the algorithm, and strong empirical results, we hope the reviewer can re-examine our paper and re-evaluate our contributions.
>
>
> [Regarding the fairness of comparison]
>
> Firstly, please note that we are making a fair comparison between MACER and Salman et al., in the sense that we are comparing **the best performance of both algorithms** (see the footnote on page 9). Salman et al. released all checkpoints and their corresponding training logs, and we compare our model with the best of their models for each $\sigma$.
>
> Secondly, we follow the reviewer's request to perform experiments on CIFAR-10 using the same number of epochs as Salman et al. used. The results can be found in Part 2.
>
> As the result shows, our models at the 150th epoch achieve comparable performance with Salman et al.'s, while they are trained almost 4x faster since our method does not rely on adversarial training. This clearly illustrates the efficiency and effectiveness of our algorithm. We have put these model checkpoints on Github for your reference.
>
>
> [Regarding other questions]
>
> - We think providing a theoretical guarantee for soft-RS is necessary. With such a theoretical guarantee, we can make sure our approximation in the algorithm aligns with the goal we try to achieve, i.e., a larger certified radius.
> - It is indeed hard to see the effect of $\gamma$ from Figure 3, and Table 11 offers a clearer demonstration.
> - Thanks for pointing out the label issue and the term issue. We have added the labels and changed the section title. Please check the updated version.
>
> We sincerely hope that the reviewer can check our response and let us know whether all your questions have been answered and all your concerns have been addressed. It would be be great if you could upgrade your rating if you are satisfied with our response. Thank you for your re-consideration.

---

> > ### Author Response · Authors · 2019-11-09
> > **Author Response (Part 2)**
> >
> >
> >   |Algorithm    | Epochs | 0.00 | 0.25 | 0.50 | 0.75 | 1.00 | 1.25 |  ACR  | Total hrs   |
> > -------------------------------------------------------------------------------------------------------
> >   |Salman-0.25  |  150   | 0.74 | 0.67 | 0.57 | 0.47 | 0.00 | 0.00 | 0.538 |    82.92    |
> >   |MACER-0.25 |  150   | 0.76 | 0.67 | 0.57 | 0.42 | 0.00 | 0.00 | 0.531 |    21.00    |
> >   |MACER-0.25 |  440   | 0.81 | 0.71 | 0.59 | 0.43 | 0.00 | 0.00 | 0.556 |    61.60    |
> > -------------------------------------------------------------------------------------------------------
> >   |Salman-0.50  |  150   | 0.50 | 0.46 | 0.44 | 0.40 | 0.38 | 0.33 | 0.709 |    82.92    |
> >   |MACER-0.50 |  150   | 0.62 | 0.57 | 0.50 | 0.44 | 0.38 | 0.29 | 0.712 |    21.00    |
> >   |MACER-0.50 |  440   | 0.66 | 0.60 | 0.53 | 0.46 | 0.38 | 0.29 | 0.726 |    61.60    |

---

> ### Author Response · Authors · 2019-11-14
> **Thanks for your attention.**
>
> Dear reviewer, we believe we have addressed your concerns and clarified your points in the rebuttal. Do you have an updated assessment (or concerns) of our paper? Thanks for your consideration.

---

### Official Review · AnonReviewer2 · 2019-10-23
**Official Blind Review #2**

**Rating:** 6

**Review:**

This paper proposes a new approach to training models robust to perturbations (or 'attacks') within an l_2 radius, by maximizing a surrogate---a soft randomized smoothing loss---for the *certified radius* (a lower bound for the l_2 attack radius) of the classifier.  This approach has the advantage of not needing to explicitly train against specific attacks, and is thus much faster and easier to optimize.  The authors provide certain theoretical guarantees and also demonstrate strong empirical results relative to two baseline approaches.

This work builds on prior work where the goal of training a classifier that is robust to attacks is phrased as maximizing the *robust radius*, the largest l_2 ball within which a data point x can be perturbed without changing the (correct) classifications of trained classifier.  Since directly maximizing this robust radius is intractable, prior work seeks to derive a lower bound which the authors term the *certified radius*.  In order to directly maximize this, the authors use randomized smoothing- in which a randomly Gaussian-smoothed version of classifier f is used to make predictions- and then motivate and develop a "soft randomized smoothing" lower bound surrogate of the certified radius to maximize, which is differentiable and provably numerically stable.

Overall, this paper is well explicated, starting with clearly written background on basic concepts and prior work, stating clear desiderata that the surrogate loss being developed should satisfy, and then providing theoretical proofs as to this.  The experiments are then thorough including core and ablation experiments to showcase the method.

One downside is that the paper does make fairly aggressive claims (e.g. "performs better than all existing provable l_2-defenses"), but then only compares to two prior / baseline approaches in the experiments.  Given the density of the field recently, this seems a bit sparse (although this reviewer is not an expert in this area)?

**Experience Assessment:**

I have read many papers in this area.

**Review Assessment: Checking Correctness Of Derivations And Theory:**

I assessed the sensibility of the derivations and theory.

**Review Assessment: Checking Correctness Of Experiments:**

I assessed the sensibility of the experiments.

**Review Assessment: Thoroughness In Paper Reading:**

I read the paper at least twice and used my best judgement in assessing the paper.

---

> ### Author Response · Authors · 2019-11-09
> **Author Response**
>
> Dear AnonReviewer2,
>
> Thanks for reading our paper thoroughly and giving us valuable advice. We are aware that there is a large body of work on provable defenses against adversarial attacks. However, as we discussed in the related work section, most previous methods can only work on a limited set of shallow networks (e.g., fully-connected ReLU networks with no more than seven layers). They cannot be scaled to deep neural networks with complex architecture (e.g., ResNet-50 for ImageNet we used in our experiments). We focus on developing a scalable and provable defense that can work on *any* network architecture. In this direction, the first method was proposed by Cohen et al. in ICML 2019. The other baseline we used (Salman et al.) achieves the previously best result, and the paper will appear in the upcoming NeurIPS 2019. Therefore, we believe that we have tracked all related baselines, and our empirical comparisons are sufficient to support our claim.
>
> We hope that this response fully addresses your concern, and we would love to answer any other questions you may have.

---

### Official Review · AnonReviewer1 · 2019-10-25
**Official Blind Review #1**

**Rating:** 8

**Review:**

This paper improves the robustness of smoothed classifiers by maximizing the certified radius, which is more efficient than adversarially train the smoothed classifier and achieves higher average robust radius and better certified robustness when the radius is not much larger than the training sigma. It proposes a novel objective which is derived by decomposing the 0/1 certified loss into the sum of 0/1 classification error and 0/1 robustness error. Three conditions are identified to make the optimization doable. Two surrogate losses (CE and hinge loss on the certified radius) for the two 0/1 errors are proposed as upper bounds of the 0/1 loss. Certified radius is derived as a function of the logits of Soft-RS to make the hinge loss differentiable. Numerical stability of the proposed objective is also analyzed by showing its gradient is bounded.

In general, the paper is well-written and the proposed objective is novel to my knowledge. I tend to accept the paper. Still, I am not sure about how much MACER improves upon the baselines, and would like to ask some questions.

1. Cross entropy is used as a surrogate for the 0/1 classification error. This is true for all cases (including all experiments in this paper) except for binary classification, where the cross entropy is less than 1 when the score on the correct class is around 0.5. It is not important but would be better if you could mention this point.

2. Have you ever tried using a tighter upper bound for the 0/1 classification error, e.g., using cross entropy loss only for the wrongly classified samples? How does it affect the results?

3. Despite showing better results, MACER seems to be using much more epochs than the two baselines (but the total hrs is smaller than (Salman et al. 2019)). Also, MACER is using a much larger k than (Cohen et al., 2019). From Figure 3 (a) we can see a larger k improves the result a lot, and from Figure 3 (b) it seems that setting lambda to a non-zero value only improves the accuracy when the radius is large. For fair comparisons, could the authors give the ACR with different values of lambda while keeping other hyper parameters unchanged? Is Salman's method still not as good when using the same number of epochs?


**Experience Assessment:**

I have published one or two papers in this area.

**Review Assessment: Checking Correctness Of Derivations And Theory:**

I assessed the sensibility of the derivations and theory.

**Review Assessment: Checking Correctness Of Experiments:**

I carefully checked the experiments.

**Review Assessment: Thoroughness In Paper Reading:**

I made a quick assessment of this paper.

---

> ### Author Response · Authors · 2019-11-09
> **Author Response**
>
> Dear AnonReviewer1,
>
> Thank you for carefully reading our paper and offering us insightful suggestions. Here are our responses to your questions and concerns:
>
> [Regarding cross entropy as a surrogate loss]
>
> This is a good catch. In fact, the logistic loss is a valid surrogate loss for binary classification (see e.g. [1]), and it only differs from the cross entropy loss by a constant ln(2) factor. Therefore, as the reviewer noted, it is not important since we can simply multiply the cross-entropy loss by a constant factor, and the surrogate condition is still satisfied in binary classification up to this constant factor.
>
> [Regarding using a tighter upper bound for the 0/1 classification error]:
>
> It is a natural idea to try to use a tighter upper bound for the 0/1 error. However, the main purpose of using surrogate losses is to overcome the non-differentiability and discontinuity of the 0/1 error. If the cross-entropy loss is only used for incorrectly classified samples, it would still be discontinuous on the decision boundary - the loss is zero for correctly classified samples, but $\geq$ 1 for misclassified ones. Hence, this approach is conceptually good but ill-posed in terms of optimization, and it indeed degrades the performance of the algorithm according to our preliminary experiments.
>
> [Regarding the experiments]
>
> (a) The number of epochs: First, please note that we are making a fair comparison between MACER and Salman et al., in the sense that we are comparing **the best performance of both algorithms** (see the footnote on page 9). Salman et al. released all checkpoints and their corresponding training logs, and we compare our model with the best of their models for each $\sigma$.
>
> Second, we follow the reviewer's request to perform experiments on CIFAR-10 using the same number of epochs as Salman et al. used. The results are attached below:
>
>   |Algorithm    | Epochs | 0.00 | 0.25 | 0.50 | 0.75 | 1.00 | 1.25 |  ACR  | Total hrs   |
> -------------------------------------------------------------------------------------------------------
>   |Salman-0.25  |  150   | 0.74 | 0.67 | 0.57 | 0.47 | 0.00 | 0.00 | 0.538 |    82.92    |
>   |MACER-0.25 |  150   | 0.76 | 0.67 | 0.57 | 0.42 | 0.00 | 0.00 | 0.531 |    21.00    |
>   |MACER-0.25 |  440   | 0.81 | 0.71 | 0.59 | 0.43 | 0.00 | 0.00 | 0.556 |    61.60    |
> -------------------------------------------------------------------------------------------------------
>   |Salman-0.50  |  150   | 0.50 | 0.46 | 0.44 | 0.40 | 0.38 | 0.33 | 0.709 |    82.92    |
>   |MACER-0.50 |  150   | 0.62 | 0.57 | 0.50 | 0.44 | 0.38 | 0.29 | 0.712 |    21.00    |
>   |MACER-0.50 |  440   | 0.66 | 0.60 | 0.53 | 0.46 | 0.38 | 0.29 | 0.726 |    61.60    |
>
> As the result shows, our models at the 150th epoch achieve comparable performance with Salman et al.'s, while they are trained almost 4x faster since our method does not rely on adversarial training. This clearly illustrates the efficiency and effectiveness of our algorithm. We have put these model checkpoints on Github for your reference.
>
> (b) The number of Gaussian samples (k): We use a larger k due to the fact that we need to estimate the certified radius in the robustness loss accurately. The inverse cdf function (Eqn 16) is sensitive to its input (i.e., the value of the probability). Using a larger number of Gaussian samples makes a better estimation of the probability and thus stabilizes the training process.
>
> (c) $\lambda$: We did provide the results about the ACRs of the model trained with different values of $\lambda$ in the appendix; please see Table 10. The table demonstrates that $\lambda$ acts as a tradeoff between accuracy and robustness. As $\lambda$ increases, accuracy at smaller radii drops, while accuracy at larger radii rises. The optimal $\lambda$ (for $\sigma$ = 0.25 and 0.50) is between 4.0 and 16.0, where accuracy and robustness are balanced.
>
> We hope our responses can address your questions and concerns about the paper. We would also be happy to answer any other questions you may have.
>
> [1] Peter L Bartlett, Michael I Jordan, and Jon D McAuliffe. Convexity, classification, and risk bounds. Journal of the American Statistical Association, 101(473):138–156, 2006

---

### Author Response · Authors · 2019-11-09
**General Author Response**

We thank all our reviewers for taking their time reading the paper and providing us with insightful comments and suggestions. We have revised our paper according to reviewers' comments and updated the new version to Open Review. We would also be glad to address any further questions or concerns from the reviewers and meta reviewers.

[Paper Updates]

- Axis labels in Figures 3 and 4 are added.
- "Ablation study" is rephrased as "Effect of hyperparameters".
- A duplicated reference is deleted.
- More model checkpoints are released for the sake of reproducibility. Link: https://github.com/MacerAuthors/macer

Thanks,
Paper470 Authors

---

### Decision · Program_Chairs · 2019-12-19

**Decision:**

Accept (Poster)

**Comment:**

The submission proposes a robustness certification technique for smoothed classifiers for a given l_2 attack radius.

Strengths:
-The majority opinion is that this work is a non-trivial extension of prior work to provide radius certification.
-The work is more efficient that strong recent baselines and provides better performance.
-It successfully achieves this while avoiding adversarial training, which is another novel aspect.

Weaknesses:
-There were some initial concerns about missing experiments and unfair comparisons but these were sufficiently addressed in the discussion.

AC shares the majority opinion and recommends acceptance.